# Phytochemical Screening and Bioactive Properties of *Juglans regia* L. Pollen

**DOI:** 10.3390/antiox11102046

**Published:** 2022-10-18

**Authors:** Natalia Żurek, Karolina Pycia, Agata Pawłowska, Ireneusz Tomasz Kapusta

**Affiliations:** Department Food Technology and Human Nutrition, Institute of Food Technology and Nutrition, University of Rzeszow, 4 Zelwerowicza St., 35-601 Rzeszow, Poland

**Keywords:** *Juglans regia*, pollen, polyphenol compounds, antioxidant activity, anticancer activity

## Abstract

Pollen is one of the major by-products of the walnut tree, yet it is poorly investigated. Thus, the aim of this study was to investigate the total phenolics, flavonoids, antioxidant, anticancer potential, and polyphenol profile of pollen obtained from male *Juglans regia* flowers. A total of 24 phenolic compounds were identified in *Juglans regia* pollen and all of them were reported for the first time for this raw material. The content of polyphenols was 408.03 mg/100 g dry weight (dw) and the most abundant components were quercetin 3-*O*-sophoroside and 4′,5,7-trihydroxy-3,6-dimethoxyflavone-7-*O*-beta-D-glucoside. The concentration of these compounds, as well as the total content of polyphenols and flavonoids, strongly determined the antioxidant and cytotoxic activity of *Juglans regia* pollen. Antioxidant action using the ABTS and CUPRAC methods had the values of 3.35 and 0.32 mmol TE/g dw, respectively. In turn, in the tests of chelating ability of ferrous ion, O_2_^•−^ and OH^−^ radical scavenging activity, of which the results were expressed as IC_50_, the values were equal to 335.01, 459.31, and 92.89 µg/mL, respectively. Among the six cancer cell lines, the strongest effect was demonstrated for Caco-2 (140.65 µg/mL) and MCF-7 (140.98 µg/mL) cells. The results provide valuable and previously unpublished data on the polyphenol composition and biological potential of *Juglans regia* pollen.

## 1. Introduction

The growing interest in polyphenolic compounds as bioactive medicinal substances of plant origin has resulted in a deeper study of many plant species that are commonly used in folk medicine [1]. Among others, this category of plants includes walnut, one of the oldest and most important trees in the world in terms of food, cosmetics, and pharmaceuticals. The walnut (*Juglans regia* L.) belongs to the walnut family (*Juglandaceae*), which includes several species of deciduous shrubs and trees. It grows in Central and Eastern Asia, North America, and Southeast Europe, where it is recognized as a traditional medicinal plant [2]. The biological tests carried out so far have confirmed the high health-promoting potential of almost all morphological parts of this plant [3].

Many of the therapeutic properties of walnut have been correlated with high contents of secondary metabolites, mainly phenolic acids, flavonoids, and naphtoquinones. The most characteristic compound for the *Juglandaceae* family is juglone (5-hydroxyl-1,4-naphthoquinone) belonging to the naphtoquinones group. Its presence was revealed in the leaves, green husk, young shoots, and fruits of the walnut [4,5,6,7]. The content of hydroxycinnamic acids (3-caffeolyquinic, 4-*p*-coumaroylquinic) and flavonoids (quercetin 3-galactoside, quercetin 3-xyloside, quercetin 3-rhamnoside) was confirmed in walnut leaves [4]. Instead, the highest concentrations of gallic acid and chlorogenic acid, and to a lesser extent-vanillic acid, caffeic acid, epicatechin, syringaldehyde, and myricetin were found in the green walnut husk [5]. Young shoots were determined to be a good source of phenolic acids (*p*-coumaric, vanillic, gallic, syringic acid), flavonoids (catechin, quercetin, myricetin), and quinones (1,4-naphthoquinone) [6]. Several bioactive ingredients were also detected in the barks of walnut (*β*-sitosterol, gallic acid, regiolone, quercetin), roots (gallic acid, catechin, quercetin), and seeds (gallic acid, chlorogenic acid, catechin, quercetin) [7,8]. Recently, much attention has been paid to polyphenolic compounds for their wide range of different functions, including antioxidant, cytotoxic, anti-inflammatory, antimicrobial, anti-diabetic, anti-aging, anti-hyperlipidemic, and hepatoprotective effects. Due to the positive impact on human health, their isolation from medicinal plants may become an effective treatment strategy for various diseases [1,9]. A wide variety and combination of techniques are used to isolate and purify the extractable phenolic compounds from a plant matrix [10].

Although many studies have already thoroughly analyzed individual morphological parts of the *J. regia* tree in terms of their phytochemical profile and health potential, the available literature lacks reports on the pollen of male walnut flowers. Walnut trees and shrubs have both male and female flowers. Male flowers are grouped into catkin-like inflorescences 5–10 cm long, and they develop laterally on the shoot in the amount of 100 to 5000 per tree [11]. Each base is made of 100–160 individual male flowers, containing 2–3 stamens. The stamen of the walnut flower consists of a short filament (1–2 mm) and a two-part anther containing approximately 6000 pollen grains. Each female cat contains on average from 11.2 to 17.6 million pollen grains, while a tree contains from 5 to 100 billion [12,13].

Considering the above, the aim of this study was to perform phytochemical screening of the polyphenol fraction isolated from the pollen of male flowers of the *J. regia* tree, together with the evaluation of antioxidant and anticancer activity by in vitro methods. Research with such a profile has not been carried out so far.

## 2. Results

### 2.1. Total Phenolic and Flavonoid Contents

Polyphenolic compounds are an important group of secondary metabolites due to their wide range of proven health benefits and high biological activity [1]. Their total content in *J. regia* pollen was assessed spectrophotometrically, and the obtained results are presented in Table 1. In the extract of *J. regia* pollen, the total content of phenolics was equal to 138.40 mg GAE/g dw, and the total content of flavonoids was 108.77 mg QE/g dw. So far, the total content of polyphenolic compounds and flavonoids has been estimated only by two authors. Okatan et al. [14] assessed the concentration of these two groups of compounds in the range of 5.05 to 11.03 mg GAE/g and 1.53 to 4.12 mg QE/g, respectively, while Cosmulescu et al. [15] studied a range between 10.80 and 17.64 mg GAE/g and 7.32 and 7.95 mg QE/g, respectively. The values estimated in our own research compared to the cited studies were several dozen times higher, which indicates a high pro-health value of *J. regia* pollen. These values are also significantly higher compared to the total phenolics and flavonoids assessed in other morphological parts of the *J*. *regia* tree. The total phenols content estimated for *J. regia* pollen was higher than the value estimated for the kernel by 2.1 times [16], green husk by 1.1 times [16], leaves by 3.0 times [4], shoot by 2.9 times [6], and stem by 1.9 times [17]. In turn, the total content of flavonoids determined for pollen was higher than the value measured for the kernel by 9.0 times [18], green kernel by 8.7 times [19], leaves by 6.9 times [4], shoot by 51.8 times [6], and stem by 16.4 times [17].

### 2.2. Antioxidant Activity

Natural antioxidants are becoming increasingly important due to their ability to prevent oxidation of other particles and to block the formation of free radicals [20]. In this study, the antioxidant capacity of the *J. regia* pollen polyphenol fraction was assessed using five chemical tests, including ABTS, O_2_^•−^, OH^−^ free radical scavenging activity, copper ion reducing capacity (CUPRAC), and iron ion chelating (ChA). As shown in Table 2, the estimated antioxidant activity using the ABTS and CUPRAC methods was 3.35 and 0.32 mmol TE/g dw, respectively. In turn, in the tests to determine the chelating ability of the ferrous ion, superoxide radical scavenging activity and hydroxyl radical scavenging activity were observed, of which the results were expressed as IC_50_, the values were equal to 335.01, 459.31, and 92.89 µg/mL, respectively.

Previously, the antioxidant activity of *J. regia* pollen was assessed using the DPPH method only [14,15]. Unfortunately, the obtained values are expressed in other units than in this paper and therefore, they cannot be compared. However, by relating the results to the antioxidant activity of other morphological parts of the *J. regia* tree, it is possible to confirm the high value of pollen. The ABTS radical scavenging activity evaluated for *J. regia* pollen was higher compared to the value determined for the kernel by 15.9 times [21], green husk by 2.7 times [22], and leaves by 2.6 times [23]. The OH^−^ radical scavenging activity was also greater than the value estimated for the walnut kernel by 3.3 times, leaves by 2.9 times, stems by 3.0 times [3], and shell by 1.1 times [24]. In turn, lower activity of *J. regia* pollen was noted in relation to the scavenging of O_2_^•−^ radicals. The obtained value was lower than the value assessed for the kernels by 1.5 times, leaves 1.6 times, stems 1.6 times [3], and shell by 2.6 times [24]. Overall, *J. regia* pollen displayed significant antioxidant potential, involving several radical scavenging mechanisms. This result suggests that the tested raw material may be of particular interest for pharmaceutical purposes or as a food fortification agent. Additionally, the correlation coefficient showed that phenolic compounds and flavonoids significantly contributed to the antioxidant activity of *J. regia* pollen (see Appendix A). The total content of phenols significantly influenced the antioxidant activity expressed as the ABTS method (r > 0.998, *p* < 0.05) and ChA method (r > −0.999, *p* < 0.01), while the total content of flavonoids influenced the activity of the CUPRAC method (r > 0.998, *p* < 0.05). Moreover, when analyzing individual polyphenolic compounds present in the highest concentration, quercetin 3-*O*-sophoroside and kaempferol 3-*O*-sophoroside were strongly correlated with the ABTS method (r > 0.998, *p* < 0.05 and r > 0.999, *p* < 0.01, respectively).

### 2.3. Effects of Cell Viability

The effect of extract from the pollen of *J. regia* flowers on the viability of cancer cells was assessed using seven cell lines, such as breast adenocarcinoma (MCF-7), colorectal adenocarcinoma (DLD-1, Caco-2), glioblastoma (U87MG), astrocytoma (U251MG), melanoma (SK-Mel-29), and human colon epithelial cells (CCD841 CoN). Cell viability with the test extract was evaluated after 24, 48, and 72 h of incubation. The results are shown in Figure 1 and Table 3. For the extract of *J. regia* pollen, cytotoxic activity was demonstrated against all cancer lines in a dose-dependent manner. The strongest cytotoxic activity was found against colorectal adenocarcinoma cells (Caco-2) 140.65 µg/mL and breast adenocarcinoma cells (MCF-7) 140.98 µg/mL, after 72 and 48 h of incubation, respectively. The highest IC_50_ values, indicating the lowest cytotoxic activity, were in turn observed for the melanoma (SK-Mel-29) and astrocytoma (U251MG) cell lines (Table 3). At the same time, the cytotoxic activity of *J. regia* pollen extract against healthy colon epithelial cells (CCD841 CoN) ranged from 549.13 to 572.12 µg/mL. These values are comparable with the results obtained for SK-Mel-29 and U251MG lines and higher for the remaining MCF-7, DLD-1, Caco-2, and U87MG cell lines.

Cytotoxic activity against healthy and neoplastic cells was evaluated for the first time for *J. regia* pollen. In our earlier work related to the cytotoxic effect of male *J. regia* flowers, we have shown the strongest activity against the MCF-7, U251MG, and U87MG cell lines. IC_50_ values for MCF-7 and U87MG lines are consistent to those obtained in this study. On the other hand, the IC_50_ values for the U251MG line after treatment with *J. regia* flower extract are nearly three times lower compared to the values assessed for the pollen [25]. However, after analyzing other existing data, it is evident that the pollens’ efficiency against some cancer lines is much stronger than that observed for other morphological parts of the walnut tree. The growth inhibition of the MCF-7 line was 3.4, 5.6, and 5.7 times higher for the pollen than for the walnut kernels, leaves [26], and the green husk [5], respectively. In turn, for the Caco-2 line, it was 1.7 and 1.2 times higher than for the kernels [27] and the leaves [28], respectively.

Most of the cited works showed that the molecular mechanisms of cell death induced by the bioactive components of the walnut tree were based on the induction of apoptosis and upregulation of Bax, downregulation of Bcl-2, and activation of caspase-3. At the same time, the cytotoxic effect depended on individual phenolic groups and/or a specific phenolic compound [29]. This study also demonstrated a strong relationship between the polyphenol composition and cytotoxic activity. As shown in Appendix A, total flavonoids content was highly correlated with activity inhibiting cell viability (MCF-7 vs. TF, r > −0.977, *p* < 0.05; MCF-7 vs. TF, r > −0.981, *p* < 0.05). There was also a significant interaction between individual phenyl compounds and cell viability (MCF-7 vs. quercetin 3-*O*-sophoroside, r > −0.999, *p* < 0.05).

### 2.4. Identification and Quantification of Phenolic Compounds

The methanol extract of the pollen of *J. regia* was purified from sugars and other highly polar compounds (e.g., organic acids, amino acids, proteins) by an initial separation using RP18 column chromatography. To evaluate the phytochemical profile of the pollen UPLC-PDA-MS, analyses were carried out using the ‘‘on-line” method. The retention times t_R_, [M-H]-, MS/MS fragments, and UV λ_max_ of the identified constituents are shown in Table 4 and the chromatogram of the pollen extract is presented in Appendix A. The analyses revealed the presence of 24 compounds. The pollen presented a composition rich in flavonoids that included 15 compounds. Their absorption maxima (λ_max_) between 250 and 270 nm and between 315 and 365 nm were typical of flavonol and flavone derivatives (Appendix A) [30]. On the other hand, the LC-MS spectra allowed us to hypothesize that they were mainly *O*-flavonol glycosides with quercetin and kaempferol as aglycones and oligosaccharides molecules linked at 3- or 7-OH position. The glycosilated flavonols contained one, two, or three sugar moieties (Appendix A). The only flavone, 4′,5,7-trihydroxy-3,6-dimethoxyflavone-7-*O*-beta-D-glucoside, was already identified in the male flowers of *J. regia* in our previous investigations [25]. The other compounds identified in the pollen extract were caffeic acid derivatives, juglanoside isomers, and spermidine esters. Caffeoyl derivatives could be distinguished by the characteristic UV spectra with two peak maximum absorption bands at 200 sh and 325–330 nm (Appendix A) and typical fragmentation pattern with daughter ions at *m*/*z* 191 and 179 (Appendix A) [31]. Juglanoside isomers are common constituents of *Juglans* species and also have shown the characteristic fragmentation pattern for this group of compounds (Appendix A) [25,32]. In turn, phenolic spermidine conjugates have frequently been found to accumulate in the reproductive organs of higher plants, particularly in the pollen grains, and they appear to determine pollen fertility [33].

The information regarding the phytochemical profile of the pollen of *J. regia* is very limited. We have identified only one report that describes the chemical composition of different pollens of walnut cultivars and our results are not completely in agreement with it, although a very similar method for polyphenol extraction has been used [14]. In the cited work, the authors analyzed pollen from walnut flowers growing in Turkey. Therefore, it should be presumed that geographic and climatic factors may have an influence on the profile and content of polyphenols. Five free phenolic acids were identified and quantified in the cited article: gallic, pyrocathecic, vanilic, caffeic, and syringic, with gallic acid being the predominant one. This discrepancy may be a result of implementation of advanced analytical techniques (UPLC-PDA-MS-MS/MS) in our study that allowed the identification of a greater number of the components, but it may also be due to the composition of the pollen itself. It is well-known that flowers’ pollen composition can vary according to geographical and environmental conditions [34].

The *J. regia* pollen was considered also from a quantitative point of view. The amounts of single compounds (mg/100 g dw) are shown in Table 4. *J. regia* pollen demonstrated a content of polyphenolics at a level of 408.03 mg/100 g dw and had quercetin 3-*O*-sophoroside as the major constituents (68.38%), followed by 4′,5,7-trihydroxy-3,6-dimethoxyflavone-7-*O*-beta-D-glucoside (6.71%) and kaempferol 3-*O*-sophoroside (6.48%). 

For comparison, in the previous works of our research group, the content of polyphenols in the male flowers of *J. regia* was 4369.73 mg/100 g dw [25], it ranged from 114.77 to 543.10 μg/g in whole walnut fruits depending on the walnut variety and the degree of maturity [35], and fluctuated between 21.20 µg/kg and 75.84 µg/kg in the dry kernels of walnuts of different cultivars [7]. In turn, in the study of Pereira [36], the quantification of the phenolics present in the different cultivar extracts of the leaves revealed the amount of these compounds ranging from ca. 65 to 73 g/kg, on a dry basis. Alternatively, walnut husk (on different sampling days) contained between 315 and 1526 mg/100 g dw of phenolics [37].

## 3. Materials and Methods

### 3.1. Materials and Reagents

The 0.25% trypsin-EDTA, 2-Deoxy-D-ribose, antibiotics (100 U/mL penicillin, and streptomycin), Dulbecco’s Modified Eagle Medium-GlutaMAX-1 (DMEM), EDTA (ethylenediaminetetraacetic acid disodium salt dihydrate), ferrozine (≥97%), fetal bovine serum (FBS), gallic acid (≥98%), NADH (β-Nicotinamide adenine dinucleotide, ≥97%), NBT (nitrotetrazolium blue chloride), PMS (phenazine methosulfate, ≥90%), neocuproine (≥98%), phosphate-buffered saline (PBS), quercetin (≥95%), ABTS (2,2′-azino-bis(3-ethylbenzothiazoline-6-sulfonic acid), and RPMI-1640 medium were purchased from Sigma-Aldrich (Steinheim, Germany). Reference standard compounds for ultra-performance liquid chromatography (UPLC) analyses were purchased Sigma-Aldrich (Darmstadt, Germany) and from Extrasynthese (Lyon, France). CellTiter 96^®^ AQueous Non-Radioactive Cell Proliferation Assay was obtained from Promega (Madison, WI, USA). Other chemicals were obtained from Chempur (Piekary Śląskie, Poland). The cell lines MCF-7, DLD-1, Caco-2 were purchased from ECACC (European Collection of Cell Cultures, Salisbury, UK). The cell line CCD841 CoN was obtained from the ATCC (American Type Culture Collection, Manassas, VA, USA). The cell lines U87MG, U251MG and SK-Mel-29 were obtained from the Nencki Institute of Experimental Biology, Polish Academy of Sciences (Warsaw, Poland).

### 3.2. Plant Material

Pollen was collected from male flowers of *Juglans regia* L. in the Subcarpathian region in Poland in May 2020. To obtain pollen, catkins were collected and frozen. After collection, the pollen was freeze-dried (ALPHA 1–2 LD plus, Osterode, Germany) and ground into a fine powder using a coffee grinder. The powdered material was stored at −20 °C until further analysis.

### 3.3. Extraction of Phenolic Compounds

The extraction and purification of polyphenols from *J. regia* pollen was performed according to our previous report [25]. The freeze-dried and powdered pollen was mixed with methanol (50%, *v*/*v*) in a 1:10 ratio and extracted by ultrasound (30 min, 30 °C, 50 Hz) (Sonic 10 ultrasonic bath, Polsonic, Poland). The solution was decanted, and the residue was extracted with methanol (80%, *v*/*v*) using ultrasound for 30 min. The resulting supernatants were combined and pre-concentrated by rotary evaporator (R-215, Buchi, Switzerland). The concentrated extract was subjected to purification on a resin LiChroprep RP-18 (40–63 µm) column. The polyphenolic fraction was eluted with methanol. The resulting extract was evaporated, freeze-dried, and stored until further analysis.

### 3.4. Determination of Total Phenolic Content (TP)

The total phenolic content was estimated using the method described by Gao et al. [38]. The extract was mixed with 2.0 mL of water, 0.2 mL of Folin-Ciocalteau reagent, and 1.0 mL Na_2_CO_3_ (20%, *w*/*v*), and left for 1 h. The absorbance was measured at the wavelength of 765 nm using a UV-VIS spectrometer (Type UV2900, Hitachi, Japan). The results were expressed as mg equivalent of gallic acid per g of dry weight (mg GAE/g dw).

### 3.5. Determination of Total Flavonoid Content (TF)

The total flavonoid content was evaluated by means of the procedure developed by Chang et al. [39]. The extract was mixed with 0.1 mL AlCl_3_ (10%, *w*/*v*), 0.1 mL sodium acetate (1 M), and 1.5 mL ethanol, 2.8 mL water, and left for 30 min. The absorbance was measured at the wavelength of 415 nm. The results were expressed in mg equivalent of quercetin per g of dry weight (mg QE/g dw).

### 3.6. Determination of Antioxidant Activity

#### 3.6.1. ABTS^+^ Scavenging Activity

The scavenging activity of pollen extract on ABTS (2,2′-azino-bis(3-ethylbenzothiazoline-6-sulfonic acid) radicals was assessed according to the method of Re et al. [40]. The extract was mixed with ABTS solution and left for 6 min. The absorbance was measured at the wavelength of 734 nm. The results were expressed as Trolox equivalent per g of dry weight (mmol TE/g dw).

#### 3.6.2. Determination of Copper Ion Reduction (CUPRAC Method)

The CUPRAC test was assessed according to the method described by Apak et al. [41]. The extract was mixed with 1.0 mL neocuproine (7.5 mM), acetate buffer (1 M, pH 7.0), and 1.0 mL copper chloride (10 mM), and left for 30 min. The absorbance was measured at the wavelength of 450 nm. The results were expressed as mmol TE/g dw.

#### 3.6.3. Chelating Ability of Ferrous Ion (ChA)

The chelating ability of iron ions was determined by the method described by Mosmann [42]. The extract was mixed with 0.4 mL ferrozine (0.25 mM) and 0.2 mL of iron sulfate (0.1 mM) and left for 10 min. The absorbance was measured at the wavelength of 562 nm. The results were expressed as IC_50_ (half-maximal inhibitory concentration).

#### 3.6.4. Superoxide Radical Scavenging Activity Assay (O_2_^•−^)

The superoxide radical scavenging activity was measured using the method described by Robak and Gryglewski [43]. The extract was mixed with 1.0 mL NBT (150 µM), 1.0 mL NADH (468 µM), and 1.0 mL PMS (60 µM), and left for 5 min. The absorbance was measured at the wavelength of 560 nm. The results were expressed as the IC_50_.

#### 3.6.5. Hydroxyl Radical Scavenging Activity Assay (OH^−^)

The hydroxyl radical scavenging activity was based on the method of Halliwell et al. [44]. The extract was mixed with 0.9 mL mixture of 2-deoxyribose (0.2 mM), EDTA (1.04 mM), iron ammonium sulphate (1.0 mM), perhydrol (0.1 M), and ascorbic acid (1.0 mM). The solution was left for 1 h, then 0.5 mL thiobarbituric acid (1%, *w*/*v*) and 1.0 mL trichloroacetic acid (2.8%, *w*/*v*) were added and it was left for 15 min. The absorbance was measured at the wavelength of 532 nm. The results were expressed as the IC_50_.

### 3.7. MTS Cell Viability Assay

Seven human cell lines, such as MCF-7 (breast adenocarcinoma), DLD-1 and Caco-2 (colon adenocarcinoma), U87MG (glioblastoma), U251MG (astrocytoma), SK-Mel-29 (melanoma), and CCD841 CoN (human colon epithelial cells) were used in this work. MCF-7, Caco-2, U87MG, U251MG, and SK-Mel-29 cell lines were cultured in DMEM media, DLD-1 and CCD841 CoN in RPMI-1640 media at 37 °C under 5% humidified atmosphere of CO_2_ in incubator (CB 170, Binder, Tuttlingen, Germany). All media were supplemented with fetal bovine serum (10%, *v*/*v*) and antibiotics (1%, *v*/*v*). For passaging, cells were washed with PBS and trypsinized (0.25% trypsin-EDTA).

Cell viability was assessed according to our previous reports [45]. Cells were seeded at a density of 8 × 10^3^ cells/well. After 24 h of incubation, cells were treated with *J. regia* pollen extracts diluted in culture media. After 24 h, 48 h, and 72 h of incubation, the MTS assay was performed according to the manufacturer’s protocol (Promega, Madison, WI, USA). Absorbance was measured at the wavelength of 490 nm with a microplate reader (SmartReader 96 Microplate Absorbance Reader, Accuris Instruments, Edison, NJ, USA). The results were expressed as the IC_50_.

### 3.8. Determination of Polyphenols Profile by UPLC-Q-TOF-MS

Polyphenolic compounds were determined using an Ultra-Performance Liquid Chromatography Array Detector (UPLC-Q-TOF-MS, Waters, Milford, MA, USA) according to the method of Żurek et al. [45]. Briefly, the separation was performed using the BEH C18 column (100 mm × 2.1 mm, 1.7 µm, Waters, Poland) at 50 °C. The injection volume of samples was 5 µL and the isocratic flow rate was 0.35 mL/min. The mobile phase consisted of solvent A (water) and solvent B (40% acetonitrile in water, *v*/*v*). The following parameters were used for TQD: capillary voltage of 3.5 kV; con voltage of 30 V; source temperature 120 °C; desolvation temperature 350 °C; con gas flow 100 L/h, and desolvation gas flow rate of 800 L/h. Polyphenolic compounds were identified and quantified based on: retention time, mass-to-charge ratio, fragmentation ions formed, and comparison with standards and literature data. All results were expressed as mg/100 g dw.

### 3.9. Statistical Analysis

All analyses were performed in triplicate (*n* = 3) and the obtained results were presented as mean and SD. Statistical analysis was performed using the Statistica 13.3 program (StatSoft, Kraków, Poland). Significant differences were assessed by Duncan’s test (*p* < 0.05) and Student’s *t*-test (*p* < 0.05; *p* < 0.01; *p* < 0.001). Additionally, the values of Pearson’s correlation coefficients were determined (*p* < 0.01; *p* < 0.05).

## 4. Conclusions

To the best of our knowledge, in this study the antioxidant properties, anticancer activity, and polyphenol profile of the fraction isolated from *J. regia* pollen were assessed for the first time. The obtained extract showed strong antioxidant properties through various reaction mechanisms, as well as a strong cytotoxic effect against the breast adenoma cell and colorectal adenocarcinoma cells line. These results are consistent with what was found in the phytochemical analyses of *J. regia* pollen, where the highest contents of quercetin 3-*O*-sophoroside, 4′,5,7-trihydroxy-3,6-dimethoxyflavone-7-*O*-beta-D-glucoside, and kaempferol 3-*O*-sophoroside were found, which allows it to have the high antioxidant capacity. In this sense, these results could indicate that the residues of *J. regia* have multiple biological effects, such as antioxidant and anticancer activity. Therefore, they could be used to design novel functional products, which could be economically exploited. Nevertheless, more research is needed to determine the bioavailability, in vivo activity, and the best approach of exploiting the potential of this raw material, taking into account both efficacy and safety, toward their possible pharmaceutical, cosmetic, or food use.

## Figures and Tables

**Figure 1 antioxidants-11-02046-f001:**
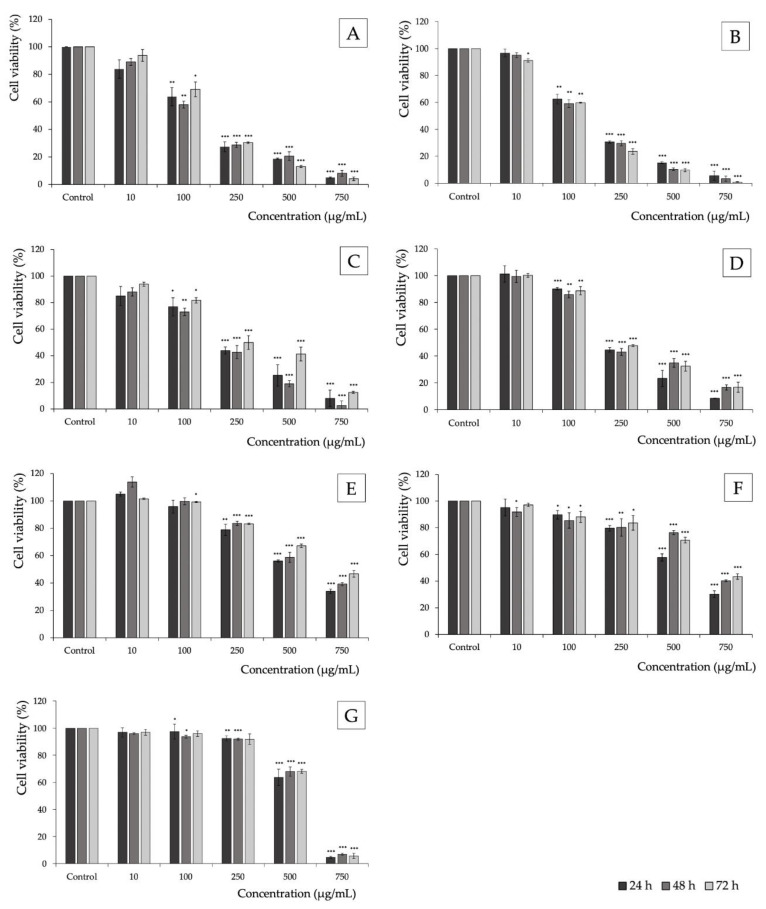
Cytotoxic effect of *J. regia* pollen extract on MCF-7 (**A**), DLD-1 (**B**), Caco-2 (**C**), U87MG (**D**), U251MG (**E**), SK-Mel-29 (**F**), and CCD841 CoN (**G**) cell lines. All cells were treated for 24, 48, and 72 h (black, gray-black, and black bars, respectively). The number of viable control (non-treated) cells at each time point served as 100% viability. Graphs represent mean values ± SD. The significance of the differences was determined by Student’s *t*-test. * *p* < 0.05, ** *p* < 0.01, *** *p* < 0.001.

**Table 1 antioxidants-11-02046-t001:** The contents of total phenolic and flavonoid of *J. regia* flowers pollen extracts.

	TP	TF
	(mg GAE/g dw)	(mg QE/g dw)
Pollen of *J. regia* flowers	138.40 ± 0.27	198.77 ± 0.20

Abbreviations: TP, total phenolic content; TF, total flavonoid content; GAE, equivalent of gallic acid; QE, equivalent of quercetin. Values are expressed as mean ± SD.

**Table 2 antioxidants-11-02046-t002:** Antioxidant activities of *J. regia* pollen extract.

	ABTS	CUPRAC	ChA	O_2_^•−^	OH^−^
	(mmol TE/g dw)		IC_50_ (µg/mL)		
Pollen of *J. regia* flowers	3.35 ± 0.03	0.32 ± 0.01	335.01 ± 5.28	459.31 ± 7.26	92.89 ± 0.04

Abbreviations: ABTS, scavenging activity assay; CUPRAC, copper ion reduction assay; ChA, chelating ability of ferrous ion; O_2_^•−^, superoxide radical scavenging activity assay; OH^−^, hydroxyl radical scavenging activity assay; TE, Trolox equivalent. Values are expressed as mean ± SD.

**Table 3 antioxidants-11-02046-t003:** IC_50_ (µg/mL) values for the examined *J. regia* pollen extract on the viability of seven human cell lines, after 24, 48, and 72 h of incubation.

No.	Cell Line	IC_50_ (µg/mL)		
		Time		
		24 h	48 h	72 h
1	MCF-7	156.73 ± 1.25	140.98 ± 4.05	174.13 ± 0.65
2	DLD-1	158.61 ± 2.49	146.54 ± 0.78	140.65 ± 3.38
3	Caco-2	222.25 ± 0.98	213.64 ± 0.56	249.76 ± 4.01
4	U87MG	232.31 ± 3.03	225.58 ± 1.30	242.12 ± 4.23
5	U251MG	568.32 ± 3.41	612.50 ± 1.11	710.51 ± 2.05
6	SK-Mel-29	570.51 ± 0.76	682.67 ± 0.93	689.65 ± 2.84
7	CCD841 CoN	549.13 ± 2.40	566.51 ± 3.09	572.12 ± 0.40

Abbreviations: MCF-7, breast adenocarcinoma cells; DLD-1, and Caco-2, colorectal adenocarcinoma cells; U87MG, glioblastoma cells; U251MG, astrocytoma cells; SK-Mel-29, melanoma cells; CCD841 CoN, colon epithelial cells. Values are expressed as mean ± SD.

**Table 4 antioxidants-11-02046-t004:** Individual phenolic compounds identified by UPLC-PDA-MS/MS in *J. regia* pollen extract.

No.	Compound	t_R_	λ_max_	[M-H]^−^ *m*/*z*		Amount
		min	nm	MS	MS/MS	mg/100 g dw
1	Quercetin 3-*O*-rutinoside-7-*O*-glucoside	2.83	255, 350	771	609, 301	0.17 ± 0.00 ^a^
2	Chlorogenic acid *	3.03	299 sh, 327	353	191, 179	0.30 ± 0.01 ^a^
3	Juglanoside B	3.43	257, 317	339	175	1.58 ± 0.03 ^abc^
4	6′-*O*-acetyl Juglanoside D	3.60	258, 348	401	355, 175	0.66 ± 0.01 ^ab^
5	6′-*O*-acetyl Juglanoside A	3.70	258, 338	369	323	0.82 ± 0.03 ^ab^
6	Juglanoside D	3.76	262, 355	355	175	0.65 ± 0.03 ^ab^
7	Quercetin 3,7-*O*-diglucoside	3.84	254, 345	625	463, 301	7.36 ± 0.09 ^f^
8	Quercetin 3-*O*-sophoroside *	3.89	255, 352	625	301	278.8 ± 1.26 ^j^
9	Quercetin 3-*O*-glucoside-xyloside	4.25	255, 352	595	301	4.95 ± 0.22 ^e^
10	*N*-caffeoyl-*N*-coumaroyl spermidine	4.30	299 sh, 312	452	289, 135	6.80 ± 0.27 ^f^
11	Kaempferol 3-*O*-sophoroside *	4.36	264, 347	609	285	26.43 ± 1.00 ^i^
12	Kaempferol 3,7-*O*-diglucoside	4.49	262, 338	609	447, 285	0.94 ± 0.01 ^ab^
13	*N,N*-dicaffeoyl spermidine	4.58	299 sh, 312	451	271	1.91 ± 0.03 ^bc^
14	Quercetin 3-*O*-glucoside *	4.64	255, 353	463	301	19.78 ± 0.31 ^h^
15	4′,5,7-Trihydroxy-3,6-dimethoxyflavone-7-*O*-beta-D-glucoside	4.79	262, 331	491	329	27.37 ± 1.73 ^i^
16	Quercetin 3-*O*-pentoside	5.05	255, 353	433	301	1.13 ± 0.03 ^ab^
17	*N,N*-dicoumarylo spermidine	5.10	297	463	273	7.49 ± 0.70 ^f^
18	Quercetin 3-*O*-pentoside	5.16	255, 353	433	301	4.09 ± 0.32 ^d^
19	Kaempferol 3-*O*-glucoside *	5.18	262, 350	447	285	2.85 ± 0.22 ^cd^
20	Quercetin 3-*O*-pentoside	5.22	255, 355	433	301	1.22 ± 0.02 ^ab^
21	Quercetin 3-*O*-rhamnoside *	5.41	255, 348	447	301	11.91 ± 0.39 ^g^
22	Kaempferol 3-*O*-rhamnoside	6.12	262, 350	431	285	0.28 ± 0.00 ^a^
23	Unidentified caffeic derivative	6.34	299 sh, 321	501	179	0.61 ± 0.02 ^ab^
24	Kaempferol 3-*O*-acetyl-glucoside	6.43	262, 338	489	447, 285	0.18 ± 0.01 ^a^
	Total					408.3 ± 7.30

Abbreviations: t_R_, retention time; UV-Vis, ultraviolet-visible; [M-H]^−^, negative ion values; *m*/*z*, mass-to-charge ratio; dw, dry weight; *, compounds identified by standards. Values are expressed as mean ± SD. The statistical significance was analyzed with Duncan’s test. Values marked with different letters indicate statistically significant differences (*p* < 0.05).

## Data Availability

Data is contained within the article and Appendix A.

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
