# Peer review of "Phytochemical Screening and Bioactive Properties of Juglans regia L. Pollen"

_antioxidants, 2022, doi:10.3390/antiox11102046_

Round 1

Reviewer 1 Report

A few comments to improve the manuscript:

1.       The review is not sure of the journal’s requirement for the order of each section (‘Results’ before ‘M&M’), otherwise, the review suggests following the normal order for each section.

2.       Line 269, mL, check throughout the manuscript.

3.       Line 347-348, at three significance levels?

4.       Line 103, is it possible to convert the same unit so that the comparison can be made?

5.       Line 113, since multiple significance levels were mentioned in M&M, which level is for the significance level here?

6.       In table 3, the pairwise comparison can be done for IC50 values for each column and the letters can be labeled for the significant differences.

Author Response

Reviewer 1

  1. The review is not sure of the journal’s requirement for the order of each section (‘Results’ before ‘M&M’), otherwise, the review suggests following the normal order for each section.

The order of individual sections is correct and complies with the editing requirements. According to the requirements, the material and methods section should follow the results and discussion.

  1. Line 269, mL, check throughout the manuscript.

It has been been corrected throughout the whole manuscript

  1. Line 347-348, at three significance levels?

Yes, this is not a mistake. Three significance levels were used, estimating the risk of error as 5%, 1% and 0.1% respectively.

  1. Line 103, is it possible to convert the same unit so that the comparison can be made?

In the opinion of the authors, it is impossible to compare the antioxidant activity for two reasons. Firstly, a different extraction protocol was used in the presented work, this applies in particular to the extraction medium. Taking into account the differences in the polyphenolic compounds polyphenolic compounds, it could significantly cause a difference in the concentration of the compounds separated from the matrix.

Secondly, the DPPH method was not used in the submitted work, but instead of ABTS. These methods use different radicals and thus different mechanisms of antioxidant reactions, which makes direct comparison impossible.

  1. Line 113, since multiple significance levels were mentioned in M&M, which level is for the significance level here?

The term „significant” in this sentence is used to refer to the meaningfull or cosiderable

  1. In table 3, the pairwise comparison can be done for IC50 values for each column and the letters can be labeled for the significant differences.

In the opinion of the authors, it is not justified to make comparisons in the columns, because different types of tumor lines have been used, and thus the mechanisms of cytotoxic action may be different, which cannot be compared

Reviewer 2 Report

The authors studied phenols and flavonoids in the extract of J. regia pollen as well as their antioxidant and anti-cancer effects. There are a few minor issues.

First, many abbreviations should be spelt out at its first instance. For example, not all readers would know “dw” stands for “dry weight” or “ABTS” stands for “2,2'-azino-bis(3-ethylbenzothiazoline-6-sulfonic acid)”. These may hinder some readers’ understanding of the work.

Second, subheadings in the Result section should be revised to avoid confusion. For example, the “effects of” should be removed from 2.1 as the sub-section is not really examining the effects of something but rather just characterizing the total amounts of phenols and flavonoids.

Third, the authors have compared the differences of phenols and flavonoids among different studies in sub-section 2.1. However, it would be better to (also) put these comparisons in Table 1, with proper citations, to make it easier to understand. Likewise, Table 2 could also be expanded to include comparisons of antioxidant activities.

Fourth, why are some compounds in Table 4 highlighted in green color? Can the authors detail which compounds were identified by comparison with commercial standards and which ones by that with literature findings?

Fifth, the claim that residues of J. regia can prevent the development of certain types of cancer seems exaggerated. Even on breast adenocarcinoma (MCF-7), the IC50 value was only about 150 mg/L (Table 3). Assuming an average molecular weight of 500 g/mol, this IC50 would be about 0.3 mM, which is far less potent than the majority of known anti-cancer drugs. Is there any in vivo evidence that such extracts do inhibit cancer development? The authors should discuss these points and revise their conclusions accordingly.

Author Response

Reviewer 2

First, many abbreviations should be spelt out at its first instance. For example, not all readers would know “dw” stands for “dry weight” or “ABTS” stands for “2,2'-azino-bis(3-ethylbenzothiazoline-6-sulfonic acid)”. These may hinder some readers’ understanding of the work.

This has corrected according to the indication.

Second, subheadings in the Result section should be revised to avoid confusion. For example, the “effects of” should be removed from 2.1 as the sub-section is not really examining the effects of something but rather just characterizing the total amounts of phenols and flavonoids.

This has been revised and updated.

Third, the authors have compared the differences of phenols and flavonoids among different studies in sub-section 2.1. However, it would be better to (also) put these comparisons in Table 1, with proper citations, to make it easier to understand. Likewise, Table 2 could also be expanded to include comparisons of antioxidant activities.

In relation to the Table 1, the total content of polyphenolic compounds and flavonoids has been estimated only by two authors. Both of the authors have studied different genotypes of the pollen of Juglans regia, and not all of these genotypes have been identified.

In regard to Table 2, we have assessed the antioxidant capacity of the J. regia pollen polyphenol fraction by five chemical tests, including ABTS, O2Ë™-, OH- free radical scavenging activity, copper ion reducing capacity (CUPRAC) and iron ion chelating (ChA) and, so far, the antioxidant activity of J. regia pollen has been assessed by means of DPPH method only. Unfortunately, the obtained values are expressed in other units than in this paper and therefore, they cannot be compared.

Thus, we belive that the update of the tables will only expand their volume without improving the readability and understanding, but will rather cause confusion for the reader.

Fourth, why are some compounds in Table 4 highlighted in green color? Can the authors detail which compounds were identified by comparison with commercial standards and which ones by that with literature findings?

The green color in Table 4 is a technical issue, we have removed it. The compounds identified by comparison with commercial standards have been marked directly in the table and the table has been described accordingly.

Fifth, the claim that residues of J. regia can prevent the development of certain types of cancer seems exaggerated. Even on breast adenocarcinoma (MCF-7), the IC50 value was only about 150 mg/L (Table 3). Assuming an average molecular weight of 500 g/mol, this IC50 would be about 0.3 mM, which is far less potent than the majority of known anti-cancer drugs. Is there any in vivo evidence that such extracts do inhibit cancer development? The authors should discuss these points and revise their conclusions accordingly.

We fully agree with this comment and we have rephrased the statement.

There is no in vivo evidence that such extracts do inhibit cancer development. We are going to determine the bioavailability, in vivo activity, and the best approach of exploiting the pollen, toward its possible pharmaceutical, cosmetic or food use.

Reviewer 3 Report

General Comments:

In this article, Å»urek et al reported their studies the antioxidant based bioactive properties of “pollen” from walnut. Pollen is indeed a major, but poorly investigated by-products of the walnut tree. And the authors investigated the total phenolics, flavonoids, antioxidant, anticancer potential and polyphenol profile of pollen obtained from male J. regia flowers. They identified 24 phenolic compounds in J. regia pollen and reported the critical content of polyphenols and other abundant components, which showed good antioxidant effects and verified from anticancer effects from in vitro experiments. They concluded that concentration of polyphenols and flavonoids strongly determined the antioxidant and cytotoxic activity of J. regia pollen. As they claimed, these results are raw and unpublished data on the polyphenol composition and biological potential of J. regia pollen for the first time. However, a discrepancy from previous report and more detailed analysis should be provided accordingly, for instance, like how the anticancer properties originated from these components and relationships between antioxidants and killing cancer cells. After the authors make these revisions, I believe their report can be considered for publication in Antioxidants.

1.      It is interesting that the authors studied the bioactive properties of “pollen” from walnut. However, is there a relationship between antioxidant effects and anticancer effects? If so, please justify it. If not, more analysis about the behind mechanism of this anticancer effects should be provided.

2.      The method of “methanol extract of the pollen of J. regia” The authors should be further illustrated other possible extracting method or reagent. Advantages or disadvantages of this method used.

3.      A discrepancy from a previous report and this paper should provide more detailed comparison and analysis about results, methods, geographical locations, etc. “We have identified only one report that describes the chemical composition of different pollens of walnut cultivars and our results are not completely in agreement with it, although very similar method for the polyphenol extraction has been used.” Page 6, Line 203-208.

4.      Still, the pollen contains many components with antioxidant properties, and which one is dominant for the strong cytotoxic effects against cancer cells? The author should also justify and analyze the behind mechanism of antioxidant agents. Still, comparison of anticancer effects with normal cell viabilities should also analyzed.  

Author Response

Reviewer 3

  1. It is interesting that the authors studied the bioactive properties of “pollen” from walnut. However, is there a relationship between antioxidant effects and anticancer effects? If so, please justify it. If not, more analysis about the behind mechanism of this anticancer effects should be provided.

The antioxidant activity in the presented manuscript was analyzed in the context of the ability to scavenge free radicals, i.e. the direct cause of cancer formation by damaging the genetic material. With this in mind, the authors' intention was to show that walnut pollen can be used as a food additive for the prevention of neoplastic diseases.

The second part of the research concerned the cytotoxic effect in relation to selected tumor lines, but in this case other mechanisms are possible, such as induction of apoptosis, etc. The presented results are only preliminary and the authors intend to explore this subject, inter alia, by explaining the mechanisms of cytotoxicity in subsequent articles.

  1. The method of “methanol extract of the pollen of J. regia” The authors should be further illustrated other possible extracting method or reagent. Advantages or disadvantages of this method used.

Similar to mentioned above. These are preliminary data. Methanol as an extractant has been used for several reasons. First, its price is low. Secondly, as shown in the literature and the authors' experience, it is well suited for the extraction of polyphenolic compounds without causing the formation of artifacts. Thirdly, it is relatively less toxic to humans compared to other organic solvents that are characterized by high volatility. Fourth, its environmental impact is also relatively low.

Thank you for this suggestion. We also think that this aspect is worth knowing and we will take it into account in further research

  1. A discrepancy from a previous report and this paper should provide more detailed comparison and analysis about results, methods, geographical locations, etc. “We have identified only one report that describes the chemical composition of different pollens of walnut cultivars and our results are not completely in agreement with it, although very similar method for the polyphenol extraction has been used.” Page 6, Line 203-208.\

Thank you for your valuable suggestion, some differences are explained in the manuscript text.

  1. Still, the pollen contains many components with antioxidant properties, and which one is dominant for the strong cytotoxic effects against cancer cells? The author should also justify and analyze the behind mechanism of antioxidant agents. Still, comparison of anticancer effects with normal cell viabilities should also analyzed.

Thank you for this valuable suggestion. We agree with this statement. As already mentioned above in the presented manuscript, we present preliminary data on the bioactive properties of walnut pollen. We are currently designing additional experiments to understand the mechanisms of cytotoxicity to neoplastic cells.

Reviewer 4 Report

This work by Å»urek et al. describes the study of the walnut tree pollen, focused on identifying major secondary metabolites and screening the antioxidant and anti-tumoural activity of the extracts.

It is very well organised and presented, very well written, and describes a sound work.

I have two minor remarks and a request for missing information to be added. Therefore, I'm recommending a major revision to give the authors the opportunity to include missing data.

Analysis data (spectra, fragmentation pathways, UV/Vis spectra) used to identify major compounds should be included as SI, together with their structures, at least for the compounds mentioned in table 4.

Minor issues

- in the abstract, please replace J. with Juglans and dw with "dry weight (dw)" on first occurrence;

- some lines in Table 4 are highlighted in green, but there’s no explanation for that highlight in the table caption or notes. 

Author Response

Reviewer 4

  1. Analysis data (spectra, fragmentation pathways, UV/Vis spectra) used to identify major compounds should be included as SI, together with their structures, at least for the compounds mentioned in table 4.

Graphic files copied from the chromatograph are included in the supplement. They contain a chromatogram and spectra UV-Vis and MS.

  1. Minnor issues

 -in the abstract, please replace J. with Juglans and dw with "dry weight (dw)" on first occurrence;

It has been corrected

-some lines in Table 4 are highlighted in green, but there’s no explanation for that highlight in the table caption or notes. 

There was a technical error here. For reasons unknown to the authors, this problem appeared while converting to a pdf file.

Reviewer 5 Report

This work was aimed to  to perform phytochemical screening of the polyphenol fraction isolated from the pollen of male flowers of the J. regia tree, together with the evaluation of antioxidant and anticancer activity by in vitro methods. Overall, this is an interesting study that shows a promise for production of Pollen. I would suggest publication after major revisions.

Specific comments are given below:

1. In Abstract, the full name of J. regia should be given in the first time.

2. Introduction: How about the novelty of the current study comparing with previous studies? Relevant information should be emphasized in the paper.

3. Antioxidant activities include many, such as DPPH, H2O2, etc. Why only do these?

4. The meaning of the statistical analysis in Figure 1 is not clearly described.

5. The format of the references needs to be harmonized in accordance with the requirements of the Journal.

6. Purity test results of Pollen should be provided.

7. It is recommended to supplement some structural assay results.

Author Response

Reviewer 5

  1. In Abstract, the full name of  regiashould be given in the first time.

This has been updated and the full name of the plant appears in the abstract.

  1. Introduction: How about the novelty of the current study comparing with previous studies? Relevant information should be emphasized in the paper.

To the best of our knowledge, in this study the anticancer activity of the fraction isolated from J. regia pollen were assessed for the first time. This is stated in the abstract (line 21), result section (line 145) and conclusions (line 354).

  1. Antioxidant activities include many, such as DPPH, H2O2, etc. Why only do these?

In this study, the antioxidant capacity of the J. regia pollen polyphenol fraction was assessed by five chemical tests, including ABTS, O2Ë™-, OH- free radical scavenging activity, copper ion reducing capacity (CUPRAC) and iron ion chelating (ChA). We think that the use of 5 different methods based on different mechanisms of action is sufficient. In detail, DPPH method was not chosen, because its mechanism of action is similar to that one of ABTS assay. In both methods the antioxidant activity of an examined antioxidant is determined in terms of relative ability to scavenge the radicals generated - DPPH· or ABTS·+.

  1. The meaning of the statistical analysis in Figure 1 is not clearly described.

In the tests for the viability of neoplastic cells, the antitumor activity of the test substance is compared with that of the control. In Figure 1, the viability of cells treated with the test extract at five concentrations was compared to the viability of untreated control. That’s why we belive that the statistical analysis in relation to Figure 1 do not require further clarifications.

  1. The format of the references needs to be harmonized in accordance with the requirements of the Journal.

This has been revised and updated.

  1. Purity test results of Pollen should be provided.

The pollen has not been contaminated as it has been removed from the flowers under the laboratory conditions. The inflorescences have been picked with closed flowers, frozen, lyophilized and then the pollen has been collected.

  1. It is recommended to supplement some structural assay results.

The supplement with the structural assay of the compounds has been added.

Round 2

Reviewer 2 Report

The authors have addressed most of my concerns. This manuscript could be accepted after more careful English language editing.

Reviewer 4 Report

I would like to thank the authors for addressing all relevant issues.

I believe the paper can be published in its present form.

Reviewer 5 Report

It can be accepted.